# The Effect of CDKN1A on the Expression of Genes Related to Milk Protein and Milk Fat Synthesis in Bovine Mammary Epithelial Cells

**DOI:** 10.3390/vetsci12060534

**Published:** 2025-06-01

**Authors:** Yuanyuan Zhang, Junxi Liang, Kai Zhang, Hong Su, Daqing Wang, Min Zhang, Feifei Zhao, Zhiwei Sun, Zhimin Wu, Guifang Cao, Yong Zhang

**Affiliations:** 1College of Veterinary Medicine, Inner Mongolia Agricultural University, Hohhot 010018, China; zyyworkaccount@163.com (Y.Z.); ljx2223128@163.com (J.L.); zhangkai040423@163.com (K.Z.); hongsu1995@126.com (H.S.); wangdaqing050789@126.com (D.W.); zhangmin5400@126.com (M.Z.); imauzff@126.com (F.Z.); 2Animal Embryo and Developmental Engineering Key Laboratory of Higher Education, Institutions of Inner Mongolia Autonomous Region, Hohhot 010018, China; 3Inner Mongolia Autonomous Region Key Laboratory of Basic Veterinary Medicine, Hohhot 010018, China; 4Tongliao Institute of Agricultural and Animal Husbandry Sciences, Tongliao 028000, China; sunzhiwei4777@163.com; 5College of Life Sciences, Inner Mongolia University, Hohhot 010021, China; wuzhimin_21@163.com

**Keywords:** bovine mammary epithelial cells, CDKN1A, milk protein synthesis, milk fat synthesis, expression vector construction

## Abstract

Through overexpression and interference of the CDKN1A gene, this study demonstrated the role of the CDKN1A gene in bovine mammary epithelial cells and analyzed the relationship between the CDKN1A gene and the synthesis of certain milk proteins and milk fat. This lays the foundation for in-depth studies on the molecular mechanisms of milk protein and milk fat synthesis in dairy cows.

## 1. Introduction

The dairy industry is an important component of animal husbandry, and the development level of the dairy cattle industry is a significant indicator of a country’s livestock sector [1]. Milk production traits, as the most important economic traits in dairy cattle, are a major focus of genetic improvement in dairy breeding [2]. However, the long generation interval and slow reproduction rate of dairy cows pose challenges. Milk production traits are both sex-limited and quantitative traits, with their phenotypes regulated by multiple microgenes. These factors limit the genetic improvement of milk production traits in dairy cattle [3]. Milk primarily contains fat, protein, lactose, inorganic salts, and vitamins. The lactating capacity of the mammary gland is mainly determined by the milk protein and milk fat synthesis capabilities of mammary epithelial cells. Unraveling the molecular mechanisms of milk protein synthesis in mammary epithelial cells will contribute to a deeper understanding of the regulatory processes of mammary gland lactation.

Milk protein is the most important nutritional component in dairy products, accounting for approximately 3% to 4% of its content. Milk protein is not only a vital source of nutrition but also possesses functions such as protecting the body, transporting substances, regulating metabolism, defending against pathogen invasion, and transmitting genetic information [4]. Milk protein also includes many bioactive components, such as antimicrobial peptides, growth factors, and immunoglobulins. The mammary gland synthesizes milk protein using free amino acids and peptide-bound amino acids extracted from the blood as raw materials [5]. Essential amino acids can enhance milk protein synthesis by promoting cell proliferation and activating the mTOR signaling pathway in bovine mammary epithelial cells [6]. Factors influencing milk protein synthesis include amino acids and energy, endocrine regulation, genetic selection, and stress responses to environmental pressures. Among these, amino acids and hormones play a significant role in regulating milk protein synthesis.

Amino acids, especially methionine, lysine, and histidine, play crucial roles in milk protein synthesis. These amino acids are often considered limiting factors in dairy cows, meaning their availability can directly impact the efficiency of milk protein production [7]. Methionine is essential for the initiation of mRNA translation and has been shown to increase protein expression, phosphorylation of STAT5a, and mTOR, which are critical for casein synthesis [8]. Studies have demonstrated that supplementing methionine can enhance milk protein yield by improving the availability of this limiting amino acid. Lysine is another limiting amino acid in dairy cow diets, particularly in corn and alfalfa silage-based diets. Supplementation with lysine has been shown to increase milk protein content and yield, highlighting its importance in milk protein synthesis. Histidine is also a limiting amino acid, especially when cows are fed grass silage-based diets. Research indicates that histidine supplementation can improve milk protein synthesis by providing the necessary substrate for protein production [9].

Milk fat synthesis in dairy cows is a complex biochemical process that primarily occurs in mammary epithelial cells, involving two pathways: de novo synthesis (synthesizing short-chain fatty acids using acetyl-CoA and malonyl-CoA) and exogenous synthesis (uptake of long-chain fatty acids from the blood). Milk fat synthesis is influenced by various factors, including diet composition, genetic factors, hormone levels, and environmental conditions. In recent years, research in genomics and metabolomics has revealed more regulatory mechanisms, providing possibilities for optimizing milk fat content and quality through technologies such as gene editing [10,11].

The CDKN1A gene, also known as p21, is a cyclin-dependent kinase inhibitor that regulates cell cycle progression by inhibiting the activity of cyclin-dependent kinases, primarily functioning during the G1 phase. In genetic studies of milk production traits in dairy cattle, polymorphisms in the CDKN1A gene have been found to be significantly associated with traits such as milk yield, milk fat percentage, and milk protein percentage [12]. The SNP, c.271C > T (rs442346530), in exon 5 of CDKN1A gene that was predicted to result in an amino acid replacement from arginine to tryptophan changed the CDKN1A protein secondary structure, suggesting CDKN1A genes play a role in milk production traits in dairy cattle and may be useful in marker-assisted breeding in dairy cattle [13]. The mTOR signaling pathway plays a central role in regulating cell growth and protein synthesis, and the CDKN1A gene may indirectly influence mammary cell function and milk production by controlling the cell cycle [14]. The synthesis of milk protein and milk fat is a complex biological process that involves the regulation of multiple genes. In recent years, with the continuous development of molecular biology techniques, an increasing number of key genes have been discovered and confirmed to participate in this process. The CSN2 gene encodes β-casein, one of the major casein proteins in milk, and its expression is positively correlated with milk protein content and milk yield. The CSN3 gene encodes κ-casein, which is responsible for maintaining the stability of casein micelles, and its expression directly affects the protein content in milk. OPN (Osteopontin) is a bioactive protein in milk and though present in low amounts, it is involved in mammary gland development and lactation and forms complexes with other functional proteins such as lactoferrin to exert synergistic effects. ACACA is the rate-limiting enzyme in de novo fatty acid synthesis, converting acetyl-CoA to malonyl-CoA, thereby influencing the synthesis of fatty acids in mammary epithelial cells and the content of milk fat. CD36 is primarily responsible for transporting free fatty acids from the blood into mammary epithelial cells, providing substrates for fat synthesis [15]. Additionally, CD36 is involved in the formation of milk fat globules. In this study, the above-mentioned genes were selected as reference genes to verify the changes in milk protein and milk fat synthesis in bovine mammary epithelial cells. siRNA binds complementarily to specific mRNA within the cell, inducing its degradation and thereby specifically silencing the expression of the target gene CDKN1A. This allows for the study of the function of specific genes at the cellular level and helps to determine the role of the CDKN1A gene in the synthesis of milk protein and milk fat in bovine mammary epithelial cells.

To clarify the effect of CDKN1A on milk protein and milk fat synthesis in bovine mammary glands, this study constructed bovine CDKN1A overexpression and interference vectors using molecular cloning technology. Using bovine mammary epithelial cells as the experimental model, the expression level of CDKN1A in the cells was altered through gene overexpression and interference methods, and changes in the expression of milk fat synthesis-related genes and milk protein in the cells were detected. This study elucidates the potential role of CDKN1A in the regulation of mammary gland lactation, providing a theoretical foundation for further elucidating the molecular mechanisms of milk component synthesis in dairy cows and improving milk quality in lactating cows through molecular biology techniques.

## 2. Materials and Methods

### 2.1. Experimental Materials

The cells used in this experiment were bovine mammary epithelial cells isolated and identified by our laboratory. The cryopreserved third-generation bovine mammary epithelial cells were thawed and cultured. The cells were seeded at a density of 1 × 10^5^ cells/mL in cell culture flasks and cultured in DMEM/F12 medium supplemented with 10% FBS and 2% antibiotics for subsequent experiments.

### 2.2. Main Reagents

BSA was purchased from Amresco in Washington, United States, fetal bovine serum (FBS) from Excell Biolog in Taicang City, Jiangsu Province, China, DMEM/F12 medium, reduced-serum medium Opti-MEM™, 0.25% trypsin, and antibiotics from Gibco in California, United States, transfection reagent Lipofectamine™ LTX from Invitrogen™ in Carlsbad, California, United States, Multisource Total RNA Miniprep Kit from AXYGEN in Union City, California, United States, total protein extraction kit from Sangon Biotech (Shanghai) Co., Ltd. in Shanghai, China, reverse transcription kit and qPCR kit from Vazyme in Nanjing, Jiangsu Province, China, high-fidelity PCR enzyme KOD Onerm PCR Master Mix from TOYOBO in Osaka, Japan, BCA protein concentration assay kit, and 5× loading buffer from Beyotime in Shanghai, China, pre-stained protein marker from Bio-rad, SDS-PAGE gel preparation kit from Solarbio in Beijing, China, CSN2 and CSN3 rabbit monoclonal antibodies from Proteintech in Wuhan, Hubei Province, China, LF, OPN, ACACA and CD36 rabbit monoclonal antibodies from Abcam in Cambridge, United Kingdom, goat anti-rabbit IgG-HRP, goat anti-mouse IgG-HRP, and mouse anti-β-actin monoclonal antibody from Tianjin Sanjian Biotechnology Company in Tianjin, China.

### 2.3. Experimental Methods

#### 2.3.1. Construction of CDKN1A Over-Expression and Interference Vectors

Total RNA was extracted from bovine mammary gland tissue using an RNA extraction kit, and the RNA concentration was measured using a BioTek Epoch 2. Subsequently, the HiScript^®^II1st Strand cDNA Synthesis kit was used for reverse transcription to obtain cDNA. The cDNA was then used as a template, and Prime STAR HS Polymerase was employed for PCR amplification of the CDKN1A gene. The PCR amplification annealing temperature was set at 65 °C, with an extension time of 30 s. The PCR primers used were as follows:

CDKN1A-HindIII-F: CCCAAGCTTGGGATGTCTGAGCTGTCCAGGGA

CDKN1A-XbaI-R: GCTCTAGAGCTTAGGGCTTCCTCTTGGAGC

The PCR-amplified product of the bovine CDKN1A gene was cloned into the HindIII and XbaI sites of the pEGFP-C1 vector. After ligation and transformation, single colonies were selected and subjected to colony PCR to identify positive colonies. The PCR reaction was performed with an annealing temperature of 65 °C and an extension time of 30 s. Positive colonies were cultured in an LB medium containing ampicillin resistance, and a portion was sent to a biotech company for sequencing. The remaining portion was mixed with 80% glycerol and stored at -80 °C. Once the sequencing results were obtained, they were compared with the target sequence. If the sequences matched, the vector construction was considered successful, and the vector was named CDKN1A-pEGFP-C1.

The siRNA for the bovine CDKN1A gene was designed and synthesized by Sangon Biotech. Three siRNAs were designed at positions 465, 423, and 453. After validation, the siRNA designed at position 423 showed the best interference effect. This siRNA was named CDKN1A-siRNA423.

#### 2.3.2. Transfection

Bovine mammary epithelial cells were seeded into 6-well plates at a density of 1 × 10^5^ cells per well. Lipofectamine™ LTX was used to transfect 5 μg of CDKN1A-C1 and CDKN1A-siRNA423 into the bovine mammary epithelial cells, respectively. The medium was replaced 24 h after transfection. After 48 h of transfection, cells were collected for RNA extraction to perform real-time quantitative PCR for detecting genes related to milk protein and milk fat synthesis. Additionally, cells were collected for protein extraction to analyze milk protein expression using Western blot. Furthermore, cells were collected for triglyceride detection.

#### 2.3.3. Real-Time Quantitative PCR

Bovine mammary epithelial cells transfected with CDKN1A overexpression and interference vectors were collected, and total RNA was extracted using an RNA extraction kit. The RNA concentration was measured using a BioTek Epoch 2, followed by reverse transcription using the HiScript^®^II1st Strand cDNA Synthesis Kit to obtain cDNA. Real-time quantitative PCR was performed using a Bio-Rad C1000 Touch Thermal Cycler. The qPCR primers are listed in Table 1. The qPCR amplification program was as follows: 95 °C for 30 s, 1 cycle; 95 °C for 5 s and 60 °C for 30 s, 40 cycles; followed by 95 °C for 15 s, 60 °C for 1 min, and 95 °C for 15 s to generate a melting curve.

#### 2.3.4. Western Blot

Total cellular proteins were extracted according to the instructions of the total protein extraction kit. The protein extracts were separated by 10% SDS-PAGE gel and then transferred onto a nitrocellulose membrane. The membrane was blocked with 5% BSA at room temperature for 4 h. Subsequently, the membrane was incubated overnight at 4 °C with primary antibodies against β-ACTIN, CSN2, CSN3, OPN, ACACA, and CD36. The next day, the membrane was washed five times with TBST and then incubated with HRP-conjugated goat anti-rabbit IgG and HRP-conjugated goat anti-mouse IgG at room temperature for 1 h. Protein bands were visualized using an enhanced chemiluminescence detection reagent, and grayscale analysis was performed using ImageJ version 1.48V.

### 2.4. Data Processing and Analysis

The results of this study were analyzed using the 2^−ΔCt^. The data were plotted using GraphPad Prism 8 and subjected to two-way ANOVA. A significance level of * *p* < 0.05 was considered statistically significant, while ** *p* < 0.01 and *** *p* < 0.001 were considered highly significant. In our study, each experiment was conducted with three biological replicates to ensure the reliability and reproducibility of the results. The data presented in the article are the mean values obtained from these replicates, along with the standard deviations to indicate the variability.

## 3. Results and Analysis

### 3.1. Construction of CDKN1A Overexpression and Interference Vectors

#### 3.1.1. Construction of CDKN1A Overexpression Vector

The electrophoresis results of the PCR-amplified CDKN1A gene product showed a target band at 486 bp (Figure 1A, see Appendix A). The PCR-amplified product of the CDKN1A gene was cloned into the HindIII and KpnI sites of the pEGFP-C1 vector. The schematic diagram of the restriction enzyme digestion is shown in Figure 1B, and the results of the PCR product and double digestion of the vector are shown in Figure 1C. After ligation and transformation, single colonies were selected for colony PCR identification. As shown in Figure 1D, the PCR product of the single colony containing the CDKN1A target fragment was 486 bp. The sequencing results of the PCR product matched the CDKN1A gene sequence (Figure 1E), indicating the successful construction of the CDKN1A overexpression vector, named pEGFP-C1-CDKN1A.

#### 3.1.2. Construction of CDKN1A Interference RNA

siRNAs were constructed by Sangon Biotech (Shanghai) Co., Ltd., with three siRNAs synthesized at positions 465, 423, and 453, respectively (Figure 2C). Cells were cultured in DMEM medium containing 10% FBS and 2% penicillin-streptomycin in a 37 °C, 5% CO_2_ incubator. Bovine mammary epithelial cells (BMECs) in the logarithmic growth phase were seeded into 6-well plates at a density of 1 × 10^5^ cells/well. After cell attachment, the cells were cultured in an antibiotic-free medium. Transfection of siRNA-NC and the three siRNA-CDKN1A constructs was performed using the Lipofectamine 3000 transfection reagent according to the manufacturer’s instructions. The medium was replaced 6 h after transfection, and the expression of CDKN1A in the cells was detected by qRT-PCR 24 h later.

### 3.2. Transfection Effect Detection of Overexpression Vector and Interference RNA

The constructed overexpression plasmid pEGFP-C1-CDKN1A was transfected into bovine mammary epithelial cells. RT-qPCR results showed that the expression of the CDKN1A gene in the transfected bovine mammary epithelial cells was significantly increased, with a statistically significant difference (*p* < 0.01, Figure 2A).

The constructed siRNA-NC, siRNA-CDKN1A465, siRNA-CDKN1A423, and siRNA-CDKN1A453 interference plasmids were transfected into bovine mammary epithelial cells. RT-qPCR results showed that the expression of the CDKN1A gene in the cells transfected with siRNA-CDKN1A423 was significantly reduced, with a statistically significant difference (*p* < 0.01, Figure 2B).

### 3.3. Effect of CDKN1A on the Expression of Genes Related to Milk Protein and Milk Fat Synthesis

The CDKN1A expression vector and interference vector were transfected into bovine mammary epithelial cells, respectively. After 48 h, RNA was extracted from the cells, and changes in the mRNA expression levels of genes related to milk protein synthesis were detected using qPCR. The results showed that after overexpression of the CDKN1A gene, the mRNA transcription levels of milk protein synthesis-related gene CSN2 were significantly decreased (*p* < 0.01) and OPN was significantly decreased (*p* < 0.001), while the mRNA transcription level of CSN3 was significantly increased (*p* < 0.001), the mRNA transcription levels of milk fat synthesis-related genes CD36 were significantly decreased (*p* < 0.001) and ACACA was significantly decreased (*p* < 0.01) (Figure 3A). After interference with the CDKN1A gene, the mRNA levels of milk protein synthesis-related gene CSN2 were significantly increased (*p* < 0.01), CSN3 and OPN were significantly increased (*p* < 0.001), the mRNA transcription levels of milk fat synthesis-related genes CD36 were significantly increased (*p* < 0.001) and ACACA was significantly increased (*p* < 0.01) (Figure 3B).

### 3.4. The Impact of CDKN1A on the Expression of Proteins Related to Milk Protein and Milk Fat Synthesis

Bovine mammary epithelial cells were transfected with CDKN1A expression vectors and interference vectors, respectively. After 48 h, cell proteins were extracted and the changes in milk protein expression levels were detected using Western blot. The results showed that overexpression of the CDKN1A gene significantly decreased the protein expression of milk protein synthesis-related genes CSN2 and CSN3 (*p* < 0.001) (Figure 4A). Conversely, after the CDKN1A gene was interfered with, the protein expression of milk protein synthesis-related genes CSN2 and CSN3 significantly increased (*p* < 0.01) (Figure 4B).

## 4. Discussion

The CDKN1A gene, also known as p21, CIP1, or WAF1, encodes a protein that inhibits cyclin-dependent kinases (CDKs), playing a critical role in regulating the mammalian cell cycle [16]. This gene is part of the CIP/KIP family, whose members inhibit the activity of CDKs and cyclins by binding to them. Although these proteins share a common cyclin-CDK inhibitory domain in their structure, they exhibit no other sequence similarities, suggesting that they may have unique functions beyond the cell cycle [17]. CDKN1A is a protein composed of 164 amino acids and contains a cyclin-dependent kinase inhibitory domain. Intrinsically disordered proteins (IDPs) and proteins with extensive disordered regions play key roles in cellular signaling, stimulus-response, and gene expression reprogramming, as their disordered nature allows them to interact flexibly with multiple partners. The presence of these disordered regions in CDKN1A implies that it may have more complex functions and interactions beyond traditional cell cycle control. Although CDKN1A is expressed in all mammalian tissues, its expression levels are relatively higher in the brain, connective tissues, and female reproductive system, particularly in smooth muscle and the extracellular matrix, where its aggregation is most pronounced [18]. At the cellular level, CDKN1A is typically found in the cytoplasm and nucleus of normal cells, except for monocytes. Given the crucial role of CDKN1A in the cell cycle, precise regulation of its activity is essential [19]. This regulation is achieved not only through direct control of its expression but also via a series of complex post-translational modifications, including phosphorylation [20], acetylation [21], methylation [22], and ubiquitination [23]. These modifications can induce various changes, such as altering the protein’s intracellular localization, enhancing its stability or promoting its degradation, and modifying its interactions with other molecules. Indeed, the anticancer effects of berberine are partly attributed to its ability to enhance the nuclear localization and stability of CDKN1A [24]. Additionally, human interaction partners of CDKN1A include proteins involved in cell cycle control, DNA damage response, and apoptosis processes.

Milk protein is the primary protein in mammalian milk, consisting mainly of two categories: casein and whey protein [25]. Casein accounts for 80% of milk protein, coagulates easily in acidic environments, and is a key component of cheese and yogurt [26]. Whey protein makes up 20%, is rich in branched-chain amino acids, is easily digestible, and is commonly used in sports nutrition products [27]. Milk protein is a high-quality source of protein, providing all essential amino acids, and supporting muscle growth, immune function, and bone health. It is widely used in the food industry as a nutritional supplement, though some individuals may have allergies or intolerances [28].

Casein CSN2 (β-casein) is an important member of the milk protein family, accounting for about 30% of bovine casein. It is encoded by the CSN2 gene, is a phosphorylated protein, and can bind to calcium ions, participating in the formation of casein micelles, which influence the texture and stability of dairy products. CSN2 is rich in essential amino acids, easily digestible, and releases bioactive peptides with antioxidant, antimicrobial, and immunomodulatory functions during digestion. In the food industry, CSN2 is widely used in products such as cheese, yogurt, and milk powder, while its genetic polymorphisms and health benefits have become research hotspots [29]. In this experiment, the mRNA transcription level and protein expression level of CSN2 were downregulated after overexpression of CDKN1A. Conversely, the mRNA transcription level and protein expression level of CSN2 were upregulated after the interference of CDKN1A. This suggests that CDKN1A inhibits the synthesis of CSN2.

Casein CSN3 (κ-casein) is another important member of the milk protein family, comprising about 10-15% of casein. It plays a critical role in the stability of milk and dairy processing [30]. Located on the surface of casein micelles, CSN3 maintains micelle stability through its hydrophilic regions, preventing precipitation. It is also the target of rennet, promoting curd formation in cheese production [31]. During digestion, CSN3 releases bioactive peptides with antimicrobial and immunomodulatory functions, and its genetic polymorphisms affect the functional properties of dairy products [32]. Through gene editing or selective breeding, the application potential of CSN3 in dairy stability and functional food development continues to expand [33]. In this experiment, overexpression of CDKN1A led to an upregulation of the mRNA transcription level of CSN3, while its protein expression level was downregulated. Conversely, the interference of CDKN1A resulted in the upregulation of both the mRNA transcription level and protein expression level of CSN2. This indicates that CDKN1A inhibits the synthesis of CSN3. The upregulation of the CSN3 mRNA transcription level after overexpression of CDKN1A might be due to prolonged transfection time, which could cause inconsistency with the protein expression level. To clarify the exact transcription level of CSN3 mRNA, it is necessary to examine additional time points.

Osteopontin (OPN) is a secretory glycoprotein with multiple biological functions, primarily exerting its effects through mechanisms such as cell adhesion, signal transduction, and metabolic regulation. OPN binds to integrin family molecules through its RGD motif within the molecule [34], or to the cell surface adhesion glycoprotein CD44 [35], thereby activating intracellular signaling pathways and mediating cell adhesion, migration, and proliferation. This cell adhesion function may provide a stable microenvironment for mammary epithelial cells, thereby influencing the efficiency of milk protein and milk fat synthesis. The mTORC1 signaling pathway is a key regulatory pathway for milk protein synthesis, and OPN may indirectly regulate the metabolic activities of mammary cells by activating related signaling pathways, thus affecting the synthesis of milk protein and milk fat [36].

ACACA is a key enzyme in fatty acid synthesis, and milk proteins (especially whey protein) may regulate ACACA activity or expression, inhibiting fatty acid synthesis and promoting fat oxidation [37]. Research indicates that branched-chain amino acids (such as leucine [38]) in whey protein may indirectly inhibit ACACA activity by activating the AMPK signaling pathway, thereby aiding in weight control, improving metabolic health, and reducing the risk of obesity-related diseases [39].

CD36 is a membrane protein involved in fatty acid uptake and inflammatory responses [40]. Whey protein and lactoferrin may regulate CD36 expression or activity, influencing fatty acid metabolism, anti-inflammatory effects, and gut health [41,42].

A study on the response of CDKN1A to DNA damage revealed that UV irradiation induces the expression of a non-coding RNA (SPUD) from the CDKN1A gene, which regulates p21 expression during the DNA damage response [43]. This suggests that environmental factors such as UV exposure can significantly impact CDKN1A expression. Another study demonstrated that RBM42, an RNA-binding protein, regulates CDKN1A splicing during DNA damage, highlighting the role of post-transcriptional mechanisms in modulating CDKN1A expression [44]. A twin cohort study investigated the genetic component of CDKN1A expression in response to ionizing radiation exposure. It was found that genetic variation accounts for 66% of the transcriptional response of CDKN1A to radiation. Specific SNPs located in transcription factor genes, such as rs205543 in the ETV6 gene and rs2287505 in the KLF7 gene, were significantly associated with CDKN1A expression levels [45]. These findings suggest that SNPs in regulatory genes can influence CDKN1A expression in real organisms.

Our study focuses on the expression of the CDKN1A gene and its potential impact on milk production. Recent research has shown that different dairy breeds exhibit significant differences in milk yield and resilience indicators, which could be influenced by genetic variations [46]. For instance, a study on milk yield variability across different breeds found that non-Holstein Friesian breeds, such as Brown Swiss and Jersey, exhibited more stable milk production with fewer severe perturbations compared to Holstein Friesian cows. This suggests that breed-specific genetic factors could play a role in milk production stability. In the context of CDKN1A expression, genetic variations such as SNPs have been shown to influence gene expression levels. For example, specific SNPs near the CDKN1A gene have been associated with altered expression in response to environmental stressors [47]. Given the observed breed differences in milk indicators, it is plausible that similar genetic variations could affect CDKN1A expression in different breeds, thereby influencing milk production.

Our study investigates the expression of the CDKN1A gene and its potential impact on milk production traits in dairy cattle. In this experiment, after overexpressing the CDKN1A gene, the mRNA transcription levels of OPN, CD36, and ACACA were downregulated to varying degrees. Conversely, after interfering with the CDKN1A gene, the mRNA transcription levels of OPN, CD36, and ACACA were upregulated to varying degrees. However, no significant changes were observed in the protein expression levels. This may be because the effects of CDKN1A on the expression levels of OPN, CD36, and ACACA have not yet manifested. Therefore, it is necessary to extend the duration of action to determine the underlying mechanisms. In summary, our study provides valuable insights into the genetic regulation of milk production traits through the CDKN1A gene. Future research should focus on validating these findings and exploring their practical applications in dairy production.

## 5. Conclusions

In this study, the CDKN1A expression vector and interference vector were successfully constructed. The results of gene overexpression and interference demonstrated that CDKN1A down-regulates the mRNA transcription levels and protein expression levels of milk protein synthesis-related genes CSN2, CSN3, OPN, ACACA, and CD36. These findings provide a theoretical foundation for further in-depth research into the molecular mechanisms by which CDKN1A influences milk protein and milk fat synthesis in dairy cows.

## Figures and Tables

**Figure 1 vetsci-12-00534-f001:**
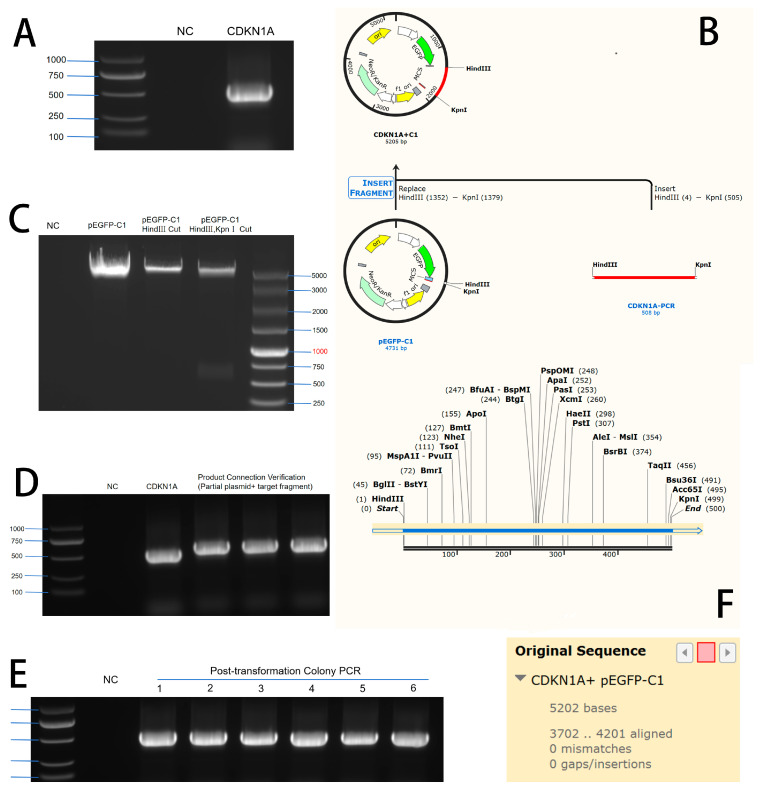
Construction of CDKN1A overexpression vector. (**A**): PCR band of the CDKN1A gene. (**B**): Schematic diagram of restriction enzyme digestion of CDKN1A and the pEGFP-C1 vector. (**C**): PCR band and digestion results of the pEGFP-C1 vector. (**D**): PCR verification after ligation. (**E**): Colony PCR verification after transformation. (**F**): Sequence alignment results of sequencing.

**Figure 2 vetsci-12-00534-f002:**
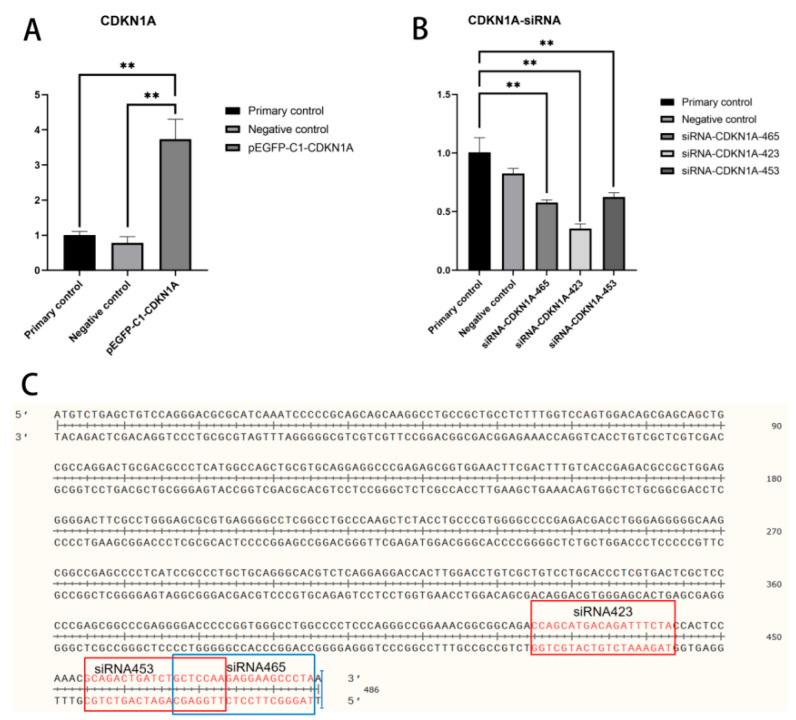
Construction and verification of siRNA of CDKN1A gene. (**A**): RT-PCR results after transfection with the CDKN1A overexpression vector. (**B**): RT-PCR results after transfection with the CDKN1A interference RNA. (**C**): Synthesis sites of the interference RNA. (** *p* < 0.01).

**Figure 3 vetsci-12-00534-f003:**
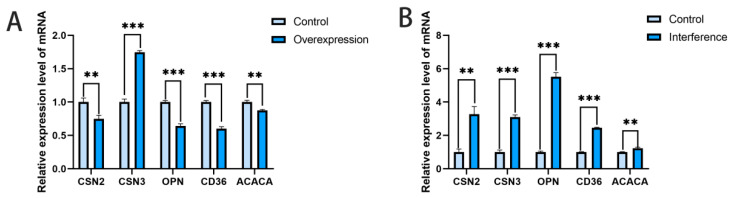
The results of qPCR detection of genes related to milk protein and milk fat synthesis. (**A**): RT-PCR results of milk protein and milk fat synthesis-related genes after transfection with the CDKN1A overexpression vector. (**B**): RT-PCR results of milk protein and milk fat synthesis-related genes after transfection with the CDKN1A interference RNA (** *p* < 0.01; *** *p* < 0.001).

**Figure 4 vetsci-12-00534-f004:**
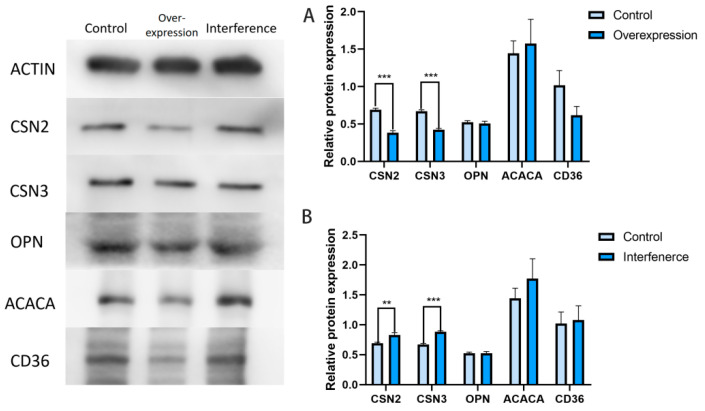
The results of Western blot detection of proteins related to milk protein and milk fat synthesis. (**A**): Western blot results of milk protein and milk fat synthesis-related genes after transfection with the CDKN1A overexpression vector. (**B**): Western-blot results of milk protein and milk fat synthesis-related genes after transfection with the CDKN1A interference RNA. (** *p* < 0.01, *** *p* < 0.001).

**Table 1 vetsci-12-00534-t001:** Information on qPCR primers for the detected genes.

Gene	Primer Sequence (5′–3′)	GenBank ID
CDKN1A	F: CCACCTGGACCTGAGCCTGAG R: CGCCGCTTGCTGTGGTAGAAG	NM_001098958.2
CSN2	F: CACAGTCTCTAGTCTATCCCTTCCC R: GGCGGCACCACCACAGG	NC_037333.1
CSN3	F: CGTCACCCACACCCACATTTATC R: TGTACTTGTAGGCTCGCCACTAG	NM_174294.2
OPN	F: TTCAGAGTCCAGATGCCACAGAG R: CTCGTCTTCTTAGGTGCGTCATG	NM_174187.2
ACACA	F: GCAGGCATCAGAAGATTATTGAAGAAG R: CGCACTCACATAACCAACCATCC	NM_174224.2
CD36	F: GAAGGCGGAAATGTTCAGAAATCAAG R: CCACACCAACACTGAGCAAGAC	NM_001278621.1
β-ACTIN	F: CCATCGGCAATGAGCGGTTC R: GGAATTGAAGGTAGTTTCGTGAATGC	NM_173979

## Data Availability

Data and materials are presented in the paper.

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
