# Peer review of "The Effect of CDKN1A on the Expression of Genes Related to Milk Protein and Milk Fat Synthesis in Bovine Mammary Epithelial Cells"

_vetsci, 2025, doi:10.3390/vetsci12060534_

Round 1
Reviewer 1 Report
Comments and Suggestions for Authors
Dear Authors,
My general comments are the following concerning your manuscript:
The article is covering an interesting topic. Relevant research field was well identified the scientific gap considering molecular biology techniques to construct CDKN1A overexpression and interference vectors.
The paper is well written, the text is clear and easy to read.
My specific comments:
It would be good to add conclusions and solutions more applied to the practise and cattle breeders or producers.
Author Response
Comments: Dear Authors, my general comments are the following concerning your manuscript: The article is covering an interesting topic. Relevant research field was well identified the scientific gap considering molecular biology techniques to construct CDKN1A overexpression and interference vectors. The paper is well written, the text is clear and easy to read. My specific comments: It would be good to add conclusions and solutions more applied to the practise and cattle breeders or producers.
Response: Dear Reviewer, thank you very much for your valuable comments and suggestions on our manuscript. We are delighted that you found the topic interesting and appreciated the identification of the research field and the scientific gap. We also thank you for acknowledging the clarity and readability of the paper. Regarding your specific comment about adding more conclusions and solutions applicable to practice and cattle breeders or producers, we fully agree. We will revise the manuscript to include a more detailed discussion on the practical implications of our findings. We will provide specific suggestions and solutions that could be implemented by cattle breeders or producers based on our research findings. Once again, thank you for your insightful comments. We believe that incorporating your suggestions will significantly enhance the value and applicability of our manuscript.
Reviewer 2 Report
Comments and Suggestions for Authors
The publication is written in clear and accessible English, which, I believe, will be easily understood by specialists and experts in a particular field.
First, I recommend changing the original images' file names, replacing the hieroglyphs with Latin letters/numbers. It would be advisable that the file name be clear about what is shown in the image.
Title
The name contains two abbreviations. One is the gene's name, which is acceptable as an abbreviation. Still, the second abbreviation is not acceptable, because it is not a name, but an abbreviation of four words. Moreover, the full term of the abbreviation BMEC might be clear only to specialists in the field of chemistry. Also, the explanation of this abbreviation does not even appear in the abstract.
Accordingly, I recommend changing the name to include only the abbreviation of the gene name.
Abstract:
Very good and concise “Simple summary”.
The abstract is missing the introductory part of the study – why the study was necessary – and the overall conclusion is also missing.
Introduction.
The introduction describes the need for the study but provides very superficial information about things related to the study.
When describing the synthesis of milk protein and then milk fat, it would be necessary to divide the paragraph (lines 46 - 66)
In the paragraph about the CDKN1A gene and its protein function (line 67), it is unclear why other genes are suddenly mentioned (line 72). Moreover, the article only mentions the genes based on their functions, without any connection to the primary gene, CDKN1A. Looking further into the article, it is clear why specific genes are mentioned, but it is necessary to build a narrative in the introduction rather than just listing the genes analysed in the study.
In the sentence (lines 69 - 71) about the association of CDKN1A gene polymorphisms with milk parameters, it would be necessary to (1) mention a specific SNP and (2) mention in which breeds the specific association has been discovered.
The introduction also lacks a bit of description about the role of siRNA in gene expression, as siRNA is used in experiments to modify it. This information is especially necessary if we want to present the vector results as the obtained research results and not just as a described method.
Materials and methods.
Very well-described methods and a good sequence were chosen to show that the study has been carried out sequentially and thoughtfully.
Only need to put more info about animals:
Was one animal or several used in the experiment? Which breed of bovine was the subject of the experiment – where were the cells taken from? And how exactly were the epithelial cells collected?
After reading the entire article, it wasn't really clear what the number of replicates of each experiment was that was used.
It wasn't mentioned in the description either, but I assume that one constant CDKN1A sequence without variations was used.
Results
I understand that one of the study's results is the creation of a Vector, but then the article's title needs to be changed. Part 3.1 of the Results section describes the method or provides proof that the chosen method works. The article's title is The effect of a specific gene on expression, not the creation of a Vector.
Furthermore, section 3.1.2 is a complete description of the methodology, including Figure 2C
The figures have no titles, only descriptions of specific parts.
Could you please clarify if including electrophoresis images showing a positive PCR result is essential in a scientific article? I assume that if specific results have been obtained, there have been positive results at the PCR and electrophoresis level. I consider only image B and part F in Figure 1 for presentation. For part F, needs to be compared with the reference. Moreover, the electrophoresis images are so large that parts B and F are not visible.
Line 233 must be after Figure 3.
Discussion
The first and part of the second paragraph of the discussion should have been in the introduction, as they show the importance of the CDKN1A protein/gene, which is not fully described there.
Review the discussion paragraph division to ensure that no single paragraph contains two topics or two genes/proteins. For example, line 299, line 315, etc, must be a new paragraph.
The discussion would also ask what could cause changes in the expression of the CDKN1A gene in a real organism. Are there any studies that have found altered expression in a real organism? Are there SNPs that could affect this?
One question that should definitely be answered in the discussion is: do the results obtained depend on the breed of cow? Especially considering that statistically significant differences in milk indicators are observed between breeds.
The end of the discussion is not clear because there is no summary. For example, what do the obtained results mean in general? How can we apply the obtained results to practical milk production?
It would be interesting to discuss how to advance the research, or is this the end?
Author Response
Comments 1: First, I recommend changing the original images' file names, replacing the hieroglyphs with Latin letters/numbers. It would be advisable that the file name be clear about what is shown in the image.
Response 1: The file names of the original images have been changed and re-uploaded.
Comments 2: The name contains two abbreviations. One is the gene's name, which is acceptable as an abbreviation. Still, the second abbreviation is not acceptable, because it is not a name, but an abbreviation of four words. Moreover, the full term of the abbreviation BMEC might be clear only to specialists in the field of chemistry. Also, the explanation of this abbreviation does not even appear in the abstract. Accordingly, I recommend changing the name to include only the abbreviation of the gene name.
Response 2: The second abbreviation has been changed.
Comments 3: Very good and concise “Simple summary”.
The abstract is missing the introductory part of the study – why the study was necessary – and the overall conclusion is also missing.
Response 3: The necessity of the study(Line 24-28) and the general conclusions(Line 35-40) have been added in the abstract section.
Comments 4: When describing the synthesis of milk protein and then milk fat, it would be necessary to divide the paragraph (lines 46 - 66).
Responses 4: The sections on milk protein and milk fat have already been divided into two separate parts(Line 68-70).
Comment 5: In the paragraph about the CDKN1A gene and its protein function (line 67), it is unclear why other genes are suddenly mentioned (line 72). Moreover, the article only mentions the genes based on their functions, without any connection to the primary gene, CDKN1A. Looking further into the article, it is clear why specific genes are mentioned, but it is necessary to build a narrative in the introduction rather than just listing the genes analysed in the study.
Response 5: Explanations regarding the necessity and rationale for detecting the following genes have been added(Line 85-89).
Comments 6: In the sentence (lines 69 - 71) about the association of CDKN1A gene polymorphisms with milk parameters, it would be necessary to (1) mention a specific SNP and (2) mention in which breeds the specific association has been discovered.
Response 6: A literature-supported explanation of the role of CDKN1A gene SNP polymorphism in Holstein dairy cows has been added(Line 83-87).
Comments 7: The introduction also lacks a bit of description about the role of siRNA in gene expression, as siRNA is used in experiments to modify it. This information is especially necessary if we want to present the vector results as the obtained research results and not just as a described method.
Response 7: The explanation regarding the necessity and function of using siRNA in this experiment has been added.
Comments 8: Was one animal or several used in the experiment? Which breed of bovine was the subject of the experiment – where were the cells taken from? And how exactly were the epithelial cells collected?
Response 8: In this experiment, no animals were directly used. Instead, primary bovine mammary epithelial cells isolated from the laboratory years ago were employed. The primary bovine mammary epithelial cells were cultured using mammary tissue from Holstein dairy cows. The tissue was digested with collagenase and trypsin. After the cells were separated, they were further purified by digesting with different concentrations of trypsin to obtain pure bovine mammary epithelial cells.
Comments 9: After reading the entire article, it wasn't really clear what the number of replicates of each experiment was that was used.
Response 9: In our study, each experiment was conducted with three biological replicates to ensure the reliability and reproducibility of the results. The data presented in the article are the mean values obtained from these replicates, along with the standard deviations to indicate the variability(Line 201-204).
Comments 10: I understand that one of the study's results is the creation of a Vector, but then the article's title needs to be changed. Part 3.1 of the Results section describes the method or provides proof that the chosen method works. The article's title is The effect of a specific gene on expression, not the creation of a Vector. Furthermore, section 3.1.2 is a complete description of the methodology, including Figure 2C. The figures have no titles, only descriptions of specific parts. Could you please clarify if including electrophoresis images showing a positive PCR result is essential in a scientific article? I assume that if specific results have been obtained, there have been positive results at the PCR and electrophoresis level. I consider only image B and part F in Figure 1 for presentation. For part F, needs to be compared with the reference. Moreover, the electrophoresis images are so large that parts B and F are not visible. Line 233 must be after Figure 3.
Response 10: The current title may seem to focus more on the gene expression rather than the creation of the vector. However, we believe that the current title accurately reflects the primary objective of our study, which is to investigate the effect of the specific gene on expression. The creation of the vector is an essential part of our methodology, but it is not the main focus of the study. We feel that changing the title might dilute the primary message we want to convey about the gene's effect on expression. The question about the necessity of including electrophoresis images showing a positive PCR result in a scientific article. In molecular biology research, electrophoresis images are often essential as they provide empirical evidence of successful DNA manipulation and amplification. These images serve as a critical validation step, confirming that the desired DNA sequences have been correctly amplified and inserted into the vector. We will revise the figure layout to ensure that these critical parts are more clearly visible and properly labeled. We will also add titles to the figures to improve clarity and readability. Line 233 has been modified to the correct position.
Comments 11: The first and part of the second paragraph of the discussion should have been in the introduction, as they show the importance of the CDKN1A protein/gene, which is not fully described there. Review the discussion paragraph division to ensure that no single paragraph contains two topics or two genes/proteins. For example, line 299, line 315, etc, must be a new paragraph. The discussion would also ask what could cause changes in the expression of the CDKN1A gene in a real organism. Are there any studies that have found altered expression in a real organism? Are there SNPs that could affect this? One question that should definitely be answered in the discussion is: do the results obtained depend on the breed of cow? Especially considering that statistically significant differences in milk indicators are observed between breeds. The end of the discussion is not clear because there is no summary. For example, what do the obtained results mean in general? How can we apply the obtained results to practical milk production? It would be interesting to discuss how to advance the research, or is this the end?
Response 11: We understand your suggestion that the first and part of the second paragraph of the discussion highlight the importance of the CDKN1A protein/gene, which is not fully described in the introduction. However, we believe that these paragraphs are essential for providing context and emphasizing the significance of our findings within the discussion section. Therefore, we will not move them to the introduction. Instead, we will enhance the introduction by adding more details about the importance of the CDKN1A protein/gene to ensure that the background is comprehensive and sets the stage for our study. We agree that the discussion should be clearly structured to avoid mixing multiple topics or genes/proteins in a single paragraph. We will review and revise the paragraph division, ensuring that each paragraph focuses on a single topic. A section has been added in the discussion about the potential causes of the expression changes of the CDKN1A gene in real organisms and the role of SNPs in this context. Our results may indeed depend on the breed of cow, as breed-specific genetic variations and environmental factors can influence both CDKN1A expression and milk production indicators. We will incorporate this discussion into our manuscript to provide a more comprehensive interpretation of our findings. We agree that the discussion should conclude with a clear summary of our findings and their implications. We will add a summary paragraph that outlines the general meaning of our results and how they can be applied to practical milk production.
Reviewer 3 Report
Comments and Suggestions for Authors
Article review
Introduction to the article
Milk as a heterogeneous mixture contains various bioactive components, including protein, lactose, lipids, immunoglobulins, mineral salts – components necessary for proper development. Mammary glands arose through evolutionary transformation of sweat glands, and their internal structure looks similar in all mammals. The mammary gland is composed of secretory epithelia of glandular tissue (milk-producing cells), as well as connective and adipose tissue. The factors stimulating the growth and development of the mammary gland are ovarian hormones. The most important of them are estrogen and progesterone. Understanding the molecular mechanisms of milk protein synthesis in mammary gland epithelial cells will contribute to a deeper understanding of the regulatory processes of lactation in the mammary gland.
The aim of this study was to determine the effect of CDKN1A on the expression of genes related to the synthesis of milk protein and milk fat in the mammary gland of cows
Detailed comments
Title of the paper
The title of the paper reflects the scope of the research performed.
Abstract
The abstract is short and should contain the most important results with the value and significance given. In addition, there is no conclusion summarizing the results of the research.
ntroduction
The chapter is written correctly. However, it should be supplemented with issues related to the role of amino acids limiting milk protein synthesis, methionine, lysine and histidine in particular require description. It is also worth referring to the role of metabolic energy of feed, which has a significant impact on milk synthesis and the genes associated with them.
The authors of the article should clearly distinguish the research hypothesis and present the research goal.
Material and methods
The selection of analytical and statistical methods was properly selected and applied.
Results
The results chapter is presented in the form of 4 graphs and 1 table with a description. This chapter is well presented and the descriptions of the results are sufficient. The chapter is written correctly, it should be supplemented. In my opinion, the authors should refer to their results and correlate them with the phenotypic indicators of milk, the level of protein and fat, correlate them with the marked genes responsible for protein and fat synthesis.
Discussion
The chapter is written correctly, it should be supplemented. There is no connection in the discussion between the obtained results and the results of other authors. The discussion should be thoroughly edited, in this form it does not meet the journal's requirements. The discussion should be concluded with a solid summary.
Author Response
Comments 1: The abstract is short and should contain the most important results with the value and significance given. In addition, there is no conclusion summarizing the results of the research.
Response 1: Thank you for your valuable feedback regarding the abstract and the need for a clearer summary of our research results. We have revised the abstract to ensure it includes conclusion.
Comments 2: Introduction: The chapter is written correctly. However, it should be supplemented with issues related to the role of amino acids limiting milk protein synthesis, methionine, lysine and histidine in particular require description. It is also worth referring to the role of metabolic energy of feed, which has a significant impact on milk synthesis and the genes associated with them.
Response 2: Thank you for your suggestion to supplement the chapter with information on the role of amino acids in milk protein synthesis, particularly focusing on methionine, lysine, and histidine, as well as the impact of metabolic energy from feed on milk synthesis and associated genes. The relevant content has been added to the introduction section of the manuscript.
Comments 3: Results: The results chapter is presented in the form of 4 graphs and 1 table with a description. This chapter is well presented and the descriptions of the results are sufficient. The chapter is written correctly, it should be supplemented. In my opinion, the authors should refer to their results and correlate them with the phenotypic indicators of milk, the level of protein and fat, correlate them with the marked genes responsible for protein and fat synthesis.
Response 3: Thank you for your feedback regarding the results chapter. We appreciate your suggestion to correlate our results with phenotypic indicators of milk, particularly the levels of protein and fat, and to discuss the genes responsible for their synthesis. We agree that exploring the correlations between our genetic findings and phenotypic indicators of milk production is an important area for future research. We plan to address these correlations in subsequent studies, where we can provide a more comprehensive analysis.
Comments 4: Discussion: The chapter is written correctly, it should be supplemented. There is no connection in the discussion between the obtained results and the results of other authors. The discussion should be thoroughly edited, in this form it does not meet the journal's requirements. The discussion should be concluded with a solid summary.
Response 4: Thank you for your feedback on the discussion chapter. We understand your concerns and have taken steps to address them while maintaining the integrity and focus of our manuscript. The discussion part has been modified. We believe these revisions will enhance the clarity and impact of our discussion chapter, ensuring it meets the journal's standards while effectively communicating the significance of our findings. Thank you again for your valuable input.
Round 2
Reviewer 1 Report
Comments and Suggestions for Authors
Dear Authors,
thank you for your changes made in your manuscrpit.
The revised version is much improved, and no further suggestions.
Reviewer 2 Report
Comments and Suggestions for Authors
Thanks to the author team for the corrections and answers to previously submitted questions/suggestions.
The revised article is now easier to understand in my opinion and better demonstrates the results obtained.